# The Effects and Mechanisms of Cyanidin-3-Glucoside and Its Phenolic Metabolites in Maintaining Intestinal Integrity

**DOI:** 10.3390/antiox8100479

**Published:** 2019-10-12

**Authors:** Jijun Tan, Yanli Li, De-Xing Hou, Shusong Wu

**Affiliations:** 1Hunan Collaborative Innovation Center for Utilization of Botanical Functional Ingredients, College of Animal Science and Technology, Hunan Agricultural University, Changsha 410128, China; jijun995@outlook.com (J.T.); liyanli125@hotmail.com (Y.L.); 2The United Graduate School of Agricultural Sciences, Faculty of Agriculture, Kagoshima University, Kagoshima 890-0065, Japan; hou@chem.agri.kagoshima-u.ac.jp

**Keywords:** cyanidin-3-glucoside, phenolic metabolites, gut microbiota, signaling pathways, intestinal injury

## Abstract

Cyanidin-3-glucoside (C3G) is a well-known natural anthocyanin and possesses antioxidant and anti-inflammatory properties. The catabolism of C3G in the gastrointestinal tract could produce bioactive phenolic metabolites, such as protocatechuic acid, phloroglucinaldehyde, vanillic acid, and ferulic acid, which enhance C3G bioavailability and contribute to both mucosal barrier and microbiota. To get an overview of the function and mechanisms of C3G and its phenolic metabolites, we review the accumulated data of the absorption and catabolism of C3G in the gastrointestine, and attempt to give crosstalk between the phenolic metabolites, gut microbiota, and mucosal innate immune signaling pathways.

## 1. Introduction

Anthocyanins belong to polyphenols, which are one kind of secondary metabolite with polyphenolic structure widely occurring in plants. They serve as key antioxidants and pigments that contribute to the coloration of flowers and fruits. Although anthocyanins vary in different plants, six anthocyanidins, including pelargonidins, cyanidins, delphinidins, peonidins, petunidins, and malvidins, are considered as the major natural anthocyanidins. Berries, such as red raspberry (*Rubus idaeus* L.), blue honeysuckle (*Lonicera caerulea* L.), and mulberry are used as folk medicine traditionally, and their extracts have been used in the treatment of disorders such as cardiovascular disease [1], obesity [2], neurodegeneration [3], liver diseases [4], and cancer [5], in recent years. Cyanidin-3-glucoside (C3G) is one of the most common anthocyanins naturally found in black rice, black bean, purple potato, and many colorful berries. C3G possesses strong antioxidant activity potentially due to the two hydroxyls on the B ring [6], as shown in Figure 1. Recent studies have suggested that C3G potentially exerts functions primarily through C3G metabolites (C3G-Ms) [7], and more than 20 kinds of C3G-Ms have been identified in serum by a pharmacokinetics study in humans [8]. Although the function and mechanism of C3G-Ms are still not clear, protocatechuic acid (PCA) [9,10,11,12], phloroglucinaldehyde (PGA) [1], vanillic acid (VA) [13,14,15], ferulic acid (FA) [16,17,18,19], and their derivates represent the main bioactive metabolites of C3G due to their antioxidant and anti-inflammatory properties. 

## 2. Absorption and Catabolism of C3G in the Gastrointestine 

Most of the anthocyanins remain stable in the stomach and upper intestine [20,21]. The stomach is considered as one of the predominant sites for anthocyanin and C3G absorption [22,23], although high concentration (85%) of anthocyanins has been found in the distal intestine [24]. There is potential for the first-pass metabolism of C3G in the stomach, that is, C3G can be effectively absorbed from the gastrointestinal tract and undergoes extensive first-pass metabolism, which can enter the systemic circulation as metabolites [25].

Anthocyanins are stable under acidic conditions but extremely unstable under alkaline conditions. The higher the pH is, the more colorless and substituent forms of anthocyanin are predominant [26]. The catabolism of C3G is mainly completed in the distal small intestine, such as ileum [22], and in the upper large intestine, such as the colon [27], with the decomposition by microbiota [28]. C3G can be hydrolyzed to their aglycones by enzymes in the small intestine, and further degraded to phenolic compounds by gut microbiota, in which microbial catabolism of C3G is performed by the cleavage of the heterocyclic flavylium ring (C-ring), followed by dehydroxylation or decarboxylation [29]. Subsequently, phase Ⅱ metabolites and multistage metabolites (including bacterial metabolites) can enter the liver and kidney to form more methylate, gluronide, and sulfate conjugated metabolites by enterohepatic circulation and blood circulation (Figure 1). 

## 3. Biological Functions of C3G-Ms 

Only several C3G-Ms have shown potential biological function, although more than 20 kinds of C3G-Ms have been identified [8,30]. PCA and phloroglucinaldehyde (PGA) are considered as the major bioactive phenolic metabolites produced by phase І metabolism, which undergo cleavage of the C ring of C3G. PCA can increase the antioxidant capacity of cells potentially by increasing the activity of antioxidant enzymes, such as catalase (CAT) in hypertensive rats or arthritis-model rats [31,32], superoxide dismutase (SOD) [33], and glutathione peroxidase (GPx) in mice or macrophages [33,34,35,36], and thus attenuate lipid peroxidation. Meanwhile, PCA has been reported to inhibit the production of inflammatory mediators, such as interleukin (IL)-6, tumor necrosis factor-α (TNF-α), IL-1β, and prostaglandin E_2_ (PGE_2_) [37,38,39], potentially by suppressing the activation of nuclear factor-κB (NF-κB) and extracellular signal-regulated kinase (ERK) [33,38] in murine BV2 microglia cells and colitis-model mice. PGA has also shown an inhibitory effect on inflammation potentially by modulating the production of IL-1β, IL-6, and IL-10 [40] in human whole blood cultures, although there are few reports about the molecular mechanisms. Our previous studies have revealed that both PCA and PGA are capable to down-regulate the MAPK pathway, especially suppress the activation of ERK, and PGA can directly bind to ERK1/2 [41] in murine macrophages.

Phase Ⅱ metabolites of C3G, such as PCA-3-glucuronide (PCA-3-Gluc), PCA-4-glucuronide (PCA-4-Gluc), PCA-3-sulfate (PCA-3-Sulf), PCA-4-sulfate (PCA-4-Sulf), VA, VA-4-sulfate (VA-4-Sulf), isovanillic acid (IVA), IVA-3-sulfate (IVA-3-Sulf), and FA, are mostly derived from PCA and PGA [1,8]. VA and FA represent the bioactive phenolic metabolites based on recent studies. VA may suppress the generation of reactive oxygen species (ROS) [42] and lipid peroxidation [32], potentially by increasing the activity of antioxidant enzymes such as SOD, CAT, and GPx [43,44], as well as the level of antioxidants such as vitamin E [43,44], vitamin C [43,44], and glutathione (GSH) [45] in mice, hamster, and diabetic hypertensive rats. Additionally, VA can inhibit the production of pro-inflammatory cytokines such as TNF-α, IL-6, IL-1β, and IL-33 by down-regulating caspase-1 and NF-κB pathways [45,46,47] in mice or mouse peritoneal macrophages and mast cells. FA has also been reported to attenuate both oxidative stress and inflammation potentially by suppressing the production of free radicals (ROS and NO in rats, rat intestinal mucosal IEC-6 cell, or murine macrophages) [48,49,50], enhancing Nrf2 expression and down-stream antioxidant enzymes (SOD and CAT in rats or swiss albino mice) [48,51], and inhibiting the activation of proinflammatory proteins (p38 and IκB in HUVEC cells) [52] and cytokines production, such as IL-18 in HUVEC cells [52], IL-1β in mice [53], IL-6 in obese rats [54], and TNF-α in mice [53]. However, both VA and FA showed a limited effect on the activation of MAPK pathway and production of inflammatory cytokines, such as monocyte chemoattractant protein-1 (MCP-1) and TNF-α in a high-fat diet-induced mouse model of nonalcoholic fatty liver disease [41]. Table 1 summarizes the biological functions of the main bioactive metabolites, including PCA, PGA, VA, and FA.

## 4. Crosstalk between Gut Microbiota and C3G&C3G-Ms

Bacteria can use phenolic compounds as substrates to obtain energy [56,57] and to form fermentable metabolites which can exert bioactive functions similar to parent anthocyanins [58], and thus, gut microbiota play an important role in the metabolism of anthocyanins and the secondary phenolic metabolites after the removal of anthocyanins’ sugar moiety [59]. 

PCA has already been proven as the gut microbiota metabolite of C3G [60], as *Lactobacillus* and *Bifidobacterium* have the maximum ability to produce the β-glucosidase so that anthocyanins are transformed to PCA [61]. *Lactobacillus* and *Bifidobacterium* are also observed to produce p-coumaric acid and FA under different carbon sources [57,62], while *Bacillus subtilis* and *Actinomycetes* are involved in the bioconversion of VA to guaiacol [63]. 

On the other hand, anthocyanins are capable of modulating the growth of special intestinal bacteria [24] and increasing microbial abundances [64]. Anthocyanins have been reported to increase the relative abundance of beneficial bacteria such as *Bifidobacterium* and *Akkermansia*, which are believed to be closely related to anti-inflammatory effects [24,65]. Monofloral honey from *Prunella Vulgaris,* rich in PCA, VA, and FA, showed protective effects against dextran sulfate sodium-induced ulcerative colitis in rats potentially through restoring the relative abundance of *Lactobacillus* [66]. Our previous studies also found that the *Lonicera caerulea* L. berry rich in C3G could attenuate inflammation potentially through the modulation of gut microbiota, especially the ratio of *Firmicutes* to *Bacteroidetes* in a mouse model of experimental non-alcoholic fatty liver disease [67]. Nevertheless, another study revealed that propolis rich in PCA, VA, and FA could suppress intestinal inflammation in a rat model of dextran sulfate sodium-induced colitis potentially by reducing the population of *Bacteroides* spp [68]. This may be because of the inhibitory and lethal effects on pathogenic bacteria by anthocyanins and their metabolites. PCA has been reported to inhibit the growth of *E. coli*, *P. aeruginosa*, and *S. aureus* [69]. VA can decrease the cucumber rhizosphere total bacterial *Pseudomonas* and *Bacillus spp.* community by changing their compositions [70]. FA is identified as highly effective against the growth of *Botrytis cinerea* isolated from grape [71]. Table 2 shows the microbial species that can biotransform C3G&C3G-Ms and the bacteriostasis effects of C3G-Ms. 

The mechanisms underlying the anti-microbial effect of anthocyanins are not clear yet. Ajiboye et al. have pointed out that PCA may induce oxidative stress in gram-negative bacteria [69], that is, PCA can combine with O_2_ to form •O^2−^, which attacks the polyunsaturated fatty acid components of the membrane to cause lipid peroxidation, and attacks the thiol group of protein to cause protein oxidation. To be more precise, •O^2−^ can be continually produced by autoxidation of PCA and semiquinone oxidation through the inhibition of NADH-quinone oxidoreductase (NQR) and succinate-quinone oxidoreductase (SQR). Although SOD converts •O^2−^ to H_2_O_2_, which can be finally changed to H_2_O and O_2_ by catalase, excessive H_2_O_2_ produces •OH during the Fenton reaction (Fe^2+^→Fe^3+^), and •OH attacks the base of DNA and results in DNA breakage. In addition, the suppression of NQR and SQR may lead to ATP depletion. Finally, bacterial death could be induced by lipid peroxidation, protein oxidation, DNA breakage, and ATP depletion. (Figure 2). 

Given these, interactions between C3G&C3G-Ms and gut microbiota can improve the bioavailability of C3G. C3G&C3G-Ms potentially ameliorate micro-ecological dysbiosis by inhibiting gram-negative bacteria. But it is worth noting that a few studies have demonstrated that the over-consumption of polyphenols had significant negative effects on reproduction and pregnancy [72,73,74]. Although it is inexplicit whether there is a correlation with the changes of gut microbiota composition, the negative effects of polyphenols-mediated modulation of gut microbiota should be focused on. 

## 5. The Potential Mechanisms of C3G&C3G-Ms against Intestinal Injury 

Multiple studies have shown that C3G&C3G-Ms have an essential role in intestinal health [55,75,76]. The potential mechanisms of C3G&C3G-Ms against intestinal injury are considered as they act in a synergistic manner between the antioxidant, anti-inflammatory, and anti-apoptosis function. 

### 5.1. Antioxidant

The protective effect of C3G&C3G-Ms against intestinal injury is largely based on their antioxidant ability. On the one hand, C3G, along with its bioactive phenolic metabolites, including PCA, VA, and FA, can up-regulate the antioxidant enzyme system, such as increasing the activities of manganese-dependent superoxide dismutase (MnSOD) [34] and GSH [34,77]. On the other hand, they can also down-regulate the pro-oxidant system, such as decrease the expression of cyclooxygenase-2 (COX-2) [77,78] and inducible nitric oxide synthase (iNOS) [77,78], and thus, decreasing the production of free radicals, including ROS [79] and reactive nitrogen species (RNS) [78]. Our previous study has shown that the *Lonicera caerulea* L. berry rich in C3G may enhance the expression of nuclear factor (erythroid-derived 2)-like 2 (Nrf2) and MnSOD during the earlier response in LPS-induced macrophages [80]. 

Nrf2 is a transcription factor with a basic leucine zipper (bZIP) that regulates the expression of antioxidant enzymes. Under normal conditions, Nrf2 is kept in ubiquitination by Cullin 3 and Kelch like-ECH-associated protein 1 (KEAP1), which facilitates ubiquitination of Nrf2. In this regard, Nrf2 forms a virtuous cycle so that it does not come into the nucleus to bind with the antioxidant response element (ARE) to modulate the transcription of down-stream genes. Once upon oxidative stress, Nrf2 can be released from KEAP1 to enter the nucleus with the disruption of cysteine residues in KEAP1 [81], or the activation of protein kinase C (PKC) [82], extracellular signal-regulated kinase (ERK) or p38 MAPKs [83], GSK-3β [84], and phosphoinositide 3-kinase (PI3K) [85]. In the nucleus, Nrf2 binds with ARE and other bZIP proteins (like small Maf) to induce down-stream genes to transcribe.

The bioactive phenolic metabolites of C3G have also been reported to activate Nrf2. PCA may increase the activities of glutathione reductase (GR) and glutathione peroxidase (GPx) by the c-Jun NH_2_-terminal kinase (JNK)-mediated Nrf2 pathway in murine macrophages, as silencing of the JNK gene expression can attenuate the PCA-induced nuclear accumulation of Nrf2 [86]. FA potentially induces the expression of Nrf2 and HO-1 via the activation of the PI3K/Akt pathway, as the specific PI3K/Akt inhibitor can suppress FA-induced Nrf2 and HO-1 expression, and block the FA-induced increase in occludin and ZO-1 protein expression in rat intestinal epithelial cells [49]. The potential mechanisms underlying the C3G-Ms induced expression of Nrf2 is summarized in Figure 3.

### 5.2. Anti-Inflammatory

Endotoxin produced by dysbacteriosis is considered as the major trigger of inflammation in intestines [87,88]. When gram-negative bacteria such as *Escherichia coli* and *Salmonella* predominate in gut, bacterial lipopolysaccharide (LPS) can form a complex called LPS binding proteins (LBP) to be associated with pattern recognition receptors (CD14) which locate on the cell membrane, and then activate toll-like receptors (TLRs), such as the TLR4 pathway, to induce inflammatory reactions in different types of cells, such as epithelial cells and immune cells [24]. TLR4 dimerizes itself and induces two major pathways, the myeloid differentiation factor 88 (MyD88)-dependent pathway and MyD88-independent pathway. In the dependent pathway, MyD88-induced phosphorylation of interleukin receptor-associated kinases 1 (IRAK1) and IRAK4 can activate the tumor necrosis-associated factor 6 (TRAF-6) adapter protein, which forms a complex with the enzymes that activate transforming growth factor beta-activated kinase-1 (TAK1) during ubiquitination. Then TAK1 induces the phosphorylation of the inhibitor kinase complex (IKKβ), which further induces the decoupling of NFκB in the dimer of NFκBp50 and NFκBp65 by degrading its inhibitory protein IκB. Finally, NFκB enters the nucleus and modulates the expression of a series of inflammatory cytokine genes [24].

Overexpression of inflammatory cytokines largely influences the expression of epithelial tight junctions (TJs) such as zonula occludens-1 (ZO-1), occludin, and claudin [89,90], which increase cellular permeability and give more access for LPS to enter cells [91,92]. The pro-inflammatory cytokines, like TNF-α, IFN-γ, and IL-1β, can induce an increase in intestinal TJ permeability potentially through the activation of myosin light chain kinase (MLCK), which appeared to be an important pathogenic mechanism contributing to the development of intestinal inflammation [93]. Another factor that aggravates intestinal inflammation is that macrophages can be recruited to adhere and infiltrate into inflammatory sites through chemokines and intercellular adhesion molecule-1 (ICAM-1), which is largely increased by the activation of the NF-κB signaling pathway [94] among several cell types including leukocytes, endothelial cells, and macrophages [95]. 

In addition to the influence on gut microbiota, the inhibitory effect of anthocyanins on epithelial inflammation is another important factor that acts against intestinal injury [64,76]. Ferrari et al. have demonstrated that the main protective effect of C3G in chronic gut inflammatory diseases is derived from the selective inhibition of the NF-κB pathway in epithelial cells [76]. Our previous studies have also shown that the *Lonicera caerulea* L. berry rich in C3G can inhibit LPS-induced inflammation potentially through TAK1-mediated mitogen-activated protein kinase (MAPK) and NF-κB pathways in an LPS-induced mouse paw edema and macrophage cell model [80]. Although the metabolites of C3G are complicated, recent studies have revealed that phenolic metabolites identified in blood circulation, such as PCA, PGA, VA, and FA, may modulate inflammatory signaling pathways. PCA, VA, and FA can suppress the production of ICAM-1, and thus, alleviate inflammatory infiltration and damage in vascular endothelial cells [52,96]. In a mouse colitis model, PCA can decrease both mRNA levels and protein concentration of Sphingosine kinases (SphK), which induces the phosphorylation of sphingosine to form sphingosine-1-phosphate (S1P), but increase the expression of S1P lyase (S1PL) which irreversibly degrade S1P, and thus, inhibit SphK/S1P pathway-mediated activation of the NF-κB pathway through S1P receptors (S1PR) [33]. The main mechanism of VA on inflammation is that it can down-regulate the MAPK pathway by suppressing the phosphorylation of ERK, JNK, and p38 [47]. It is reported that FA may prevent macrophages from responding to LPS, potentially through target myeloid differentiation factor 88 (MyD88) mediated pro-inflammatory signaling pathways [50,97], while other studies suggested that FA may increase the expression of TJs, such as occludin and ZO-1 via regulating HO-1 expression to prevent LPS enter the cells [52]. In our previous studies, both C3G and its phenolic metabolites showed inhibitory effects on LPS-activated inflammatory pathways in macrophages, and C3G can directly bind to TAK1 and ERK1/2, while PGA, one of phase I metabolites, can also directly bind to ERK1/2 [41,80]. These studies suggest that C3G and its phenolic metabolites may attenuate both a primary and secondary inflammatory response and by inactivating pro-inflammatory pathways and enhancing cellular barrier function (Figure 4).

### 5.3. Anti-Apoptosis 

Under normal conditions, the homeostasis between apoptosis and proliferation of intestinal epithelial cells regulates the normal morphological structure and physiological function of the intestinal tract [98]. However, pathological factors, such as intestinal flora disorder, may induce local inflammation and subsequently, the infiltration of immune cells, such as leukocytes, that can be easily activated by microbial products causing the overproduction of RNS and ROS, which finally causes abnormal apoptosis among intestinal cells [34,99,100]. Although mechanisms underlying apoptosis are complicated, it has been considered that apoptosis is mainly mediated by two ways [101]. On the one hand, pro-apoptotic factors such as ROS may change mitochondrial permeability and induce the release of second mitochondria-derived activator of caspases (SMACs) into the cytoplasm to bind and inactivate the inhibitor of apoptosis proteins (IAPs) like Bcl-2 [102,103], which inhibit the activation of caspase and contribute to protecting intestinal epithelial cells from apoptosis [104]. On the other hand, increased mitochondrial permeability can also cause the release of cytochrome c (Cyto C), the inducer of apoptotic protease activating factor-1 (Apaf-1), through the mitochondrial apoptosis-induced channel (MAC), which is generally suppressed by Bcl-2 family proteins [105], to induce the production of caspase 9 and caspase 3 and promote apoptosis [105]. 

The effects of C3G on apoptosis are various in different cell models. It has been reported that C3G can potentially inhibit human colon cancer cell proliferation through promoting apoptosis and suppressing angiogenesis [106,107]. But in other normal cases, C3G showed the protective effects on gastrointestinal cells, as well as endothelial cells, and obviously inhibited apoptosis by regulating apoptosis associated proteins, such as reducing the cytoplasmatic levels in Bax [108] and inhibiting the expression of caspase-8 [109], caspase-9 [108], and caspase-3 [108,109] to attenuate gastrointestinal damage. The impacts of C3G-Ms on apoptosis are similar to C3G, and multiple studies have suggested that PCA [35,36,100] and FA [110,111] may act against apoptosis in various models, although other studies revealed that C3G-Ms (PCA, FA) might promote apoptosis of colorectal adenocarcinoma cells [112]. The mechanism of C3G-Ms against apoptosis is still unclear, but a recent study has shown that in addition to the direct quenching of ROS, PCA may inhibit the expression of pro-apoptotic Bax in mitochondria and subsequently, increase the ratio of Bcl-2/Bax to reduce the production of caspase 8, caspase 9, and caspase 3 in injured gastrointestinal mucosa (Figure 5) [36]. 

## 6. Conclusions 

Due to the strong antioxidant and anti-inflammatory properties, anthocyanins present in natural products offer great hope as an alternative therapy for chronic disorders, such as cardiovascular disease, fatty liver disease, inflammatory bowel disease, and glucose-lipid metabolism disorders. Maintaining the gut integrity plays an important role in the health-promoting functions of anthocyanins, as the intestinal tract is not only the main place for digestion and absorption of food but also the first defense barrier against external pathogens and stimulus. It is commonly believed that the degradation of anthocyanins in the gastrointestinal tract decreases their bioavailability; however, recent studies based on the microbiome and metabonomics have suggested that the interaction between natural bioactive compounds and gut microbiota may potentially increase health benefits. On the one hand, anthocyanins can modulate the gut microbiota composition through either bacteriostasis effect or as nutrients to promote the growth of specific microbes. On the other hand, gut microbiota may break down anthocyanins to form multiple metabolites, which are absorbed into the systemic circulation to exert positive or negative effects. Thus, understanding the interactions between anthocyanins and microorganisms, as well as the effects of anthocyanin-derived metabolites on cellular signaling pathways, is necessary for the rational use of anthocyanins. The breakdown of C3G in the gastrointestinal tract generates a series of secondary phenolic metabolites, which take up the main part of C3G-derived bioactive phenolics in circulation. Those metabolites, such as PCA, PGA, VA, and FA, not only regulate the gut microbiota potentially by their lethal effects on microorganisms but also affect the Nrf2-mediated antioxidant system and inflammatory pathways, such as the TAK1-mediated MAPK pathway and SphK/S1P mediated NF-κB pathway. Based on this, C3G and its metabolites improve the microenvironment and attenuate the oxidative stress and inflammation to reduce the cell death of enterocytes, which ultimately maintain intestinal integrity and function. However, species-specific microbial communities and their products affected by C3G and its bioactive metabolites, and how those products regulate signaling pathways and physiological responses are still not clear. Future studies based on multi-omics analysis will provide an insight into both the health benefits and negative effects of C3G and contribute to the rational use of this common natural anthocyanin.

## Figures and Tables

**Figure 1 antioxidants-08-00479-f001:**
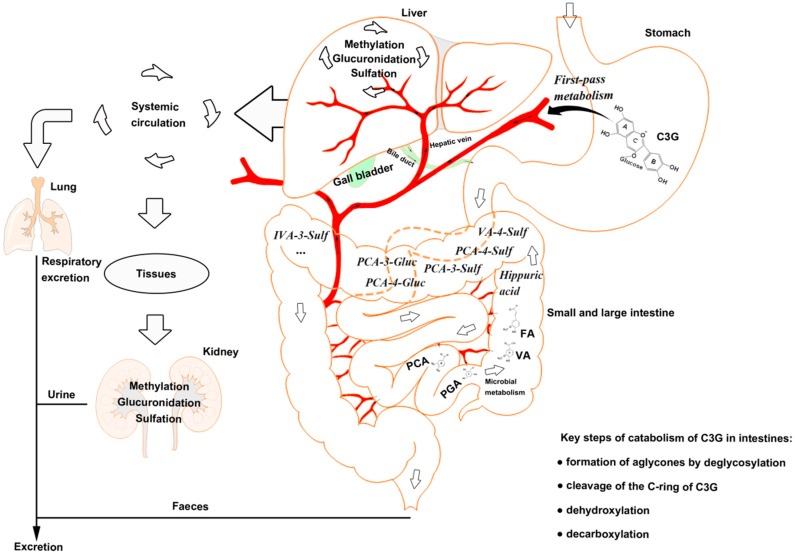
The catabolism process of cyanidin-3-glucoside (C3G) in an organism. C3G can be hydrolyzed to its aglycone by enzymes in the small intestine, and further degraded to phenolic compounds by gut microbiota. Microbial catabolism of C3G in the distal small intestine and large intestine is performed by the cleavage of the heterocyclic flavylium ring (C-ring), followed by dehydroxylation or decarboxylation to form multistage metabolites, which enter the liver and kidney by circulation. C3G, cyanidin-3-glucoside; FA, ferulic acid; PCA, protocatechuic acid; PGA, phloroglucinaldehyde; VA, vanillic acid.

**Figure 2 antioxidants-08-00479-f002:**
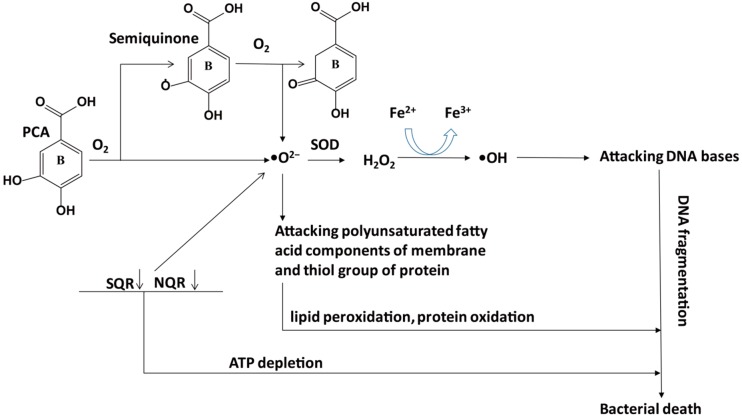
Potential mechanisms underlying the lethal effect of PCA on gram-negative bacteria. Autoxidation of PCA and semiquinone oxidation through the inhibition of NADH-quinone oxidoreductase (NQR) and succinate-quinone oxidoreductase (SQR) can cause ATP depletion and produce •O^2−^, which attacks the polyunsaturated fatty acid components of the membrane to cause lipid peroxidation and attacks the thiol group of protein to cause protein oxidation. Although SOD converts •O^2-^ to H_2_O_2_, which can be finally changed to H_2_O and O_2_ by catalase, excessive H_2_O_2_ produces •OH during the Fenton reaction (Fe^2+^→Fe^3+^), and •OH attacks DNA bases to cause DNA fragmentation. Ultimately, lipid peroxidation, protein oxidation, DNA fragmentation, and ATP depletion induce bacterial death. PCA, protocatechuic acid; SOD, superoxide dismutase.

**Figure 3 antioxidants-08-00479-f003:**
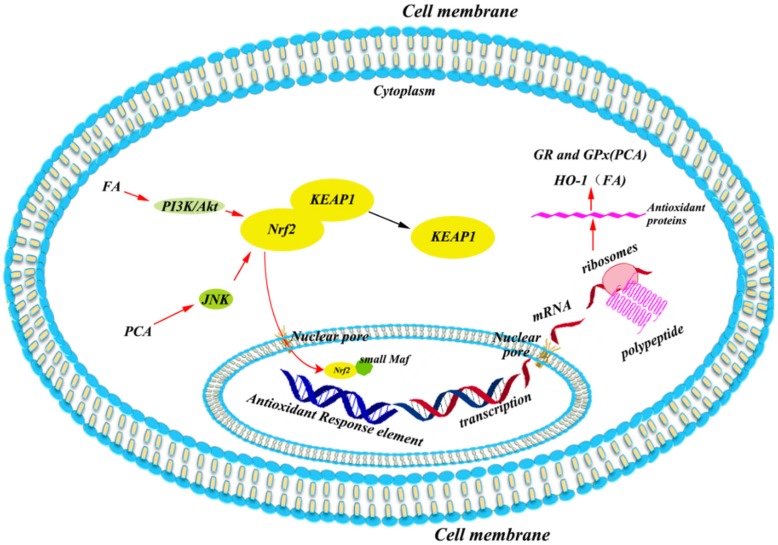
Potential mechanisms underlying the C3G-Ms regulated Nrf2 system. PCA and FA may induce the nuclear translocation of Nrf2 via JNK and PI3K/Akt pathways, respectively. FA, ferulic acid; GPx, glutathione peroxidase; GR, glutathione reductase; JNK, c-Jun NH_2_-terminal kinase; KEAP1, Kelch like-ECH-associated protein 1; Nrf2, nuclear factor (erythroid-derived 2)-like 2; PCA, protocatechuic acid; PI3K, phosphatidylinositol 3-Kinase.

**Figure 4 antioxidants-08-00479-f004:**
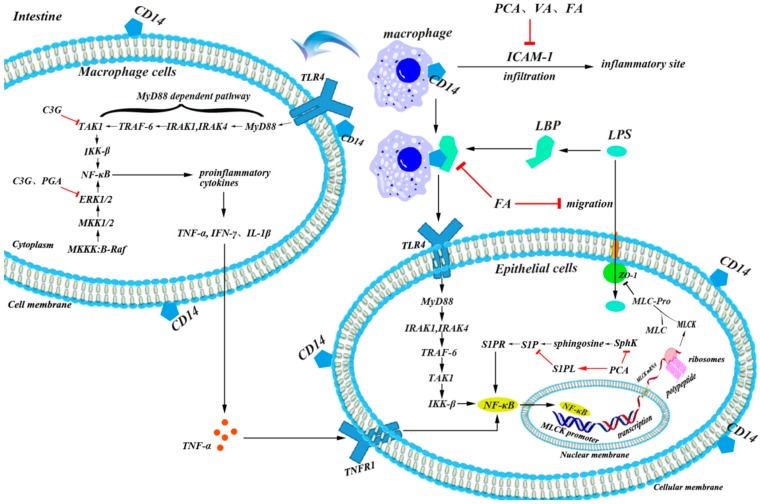
Potential mechanisms of C3G&C3G-Ms in attenuating intestinal inflammation. C3G and its phenolic metabolites mainly modulate inflammation by three ways, first, to suppress the production of chemotactic factors such as ICAM-1 and thus alleviate inflammatory infiltration, second, to down-regulate inflammatory pathways such as TAK1-mediated MAPK pathway and SphK/S1P mediated NF-κB pathway, finally, the down-regulated inflammatory pathways, and up-regulated antioxidant pathway, as mentioned in Figure 3, will maintain sufficient expression of tight junction proteins such as ZO-1 to promote normal intestinal barrier function, and thus prevent LPS from entering mucosal cells. C3G, cyanidin 3-glucoside; FA, ferulic acid; ICAM-1, intercellular adhesion molecule-1; PCA, protocatechuic acid; PGA, phloroglucinaldehyde; SphK, Sphingosine kinases; S1P, sphingosine-1-phosphate; TAK1, transforming growth factor beta-activated kinase-1; VA, vanillic acid.

**Figure 5 antioxidants-08-00479-f005:**
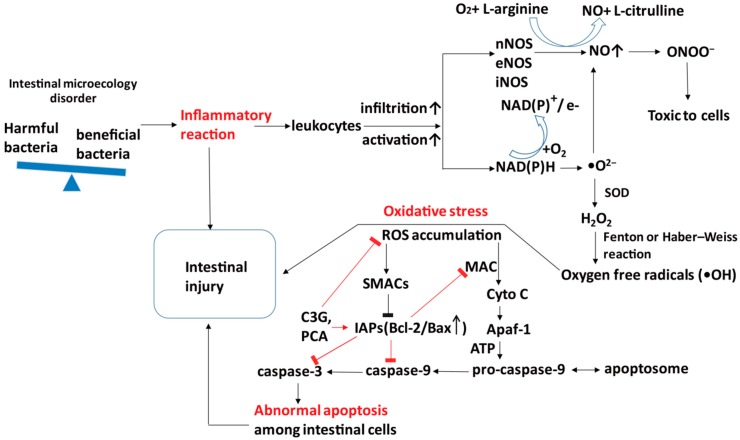
Potential mechanisms of C3G&C3G-Ms against apoptosis in intestinal epithelial cells. Intestinal flora disorder can induce the overproduction of pro-apoptotic factors such as ROS to increase mitochondrial permeability and cause the release of SMACs to bind and inactivate IAPs, such as Bcl-2. Since IAPs inhibit the activation of MAC and caspase to inhibit apoptosis, the inactivation of IAPs will induce the release of Cyto C through MAC, and subsequently induce the expression of Apaf-1 and caspase to cause apoptosis. C3G and its metabolites PCA can directly quench ROS and activate IAPs to inhibit the release of Cyto C and expression of caspases. Apaf-1, apoptotic protease activating factor-1; Cyto C, cytochrome C; C3G, cyanidin 3-glucoside; IAPs, inhibitor of apoptosis proteins; MAC, mitochondrial apoptosis-induced channel; PCA, protocatechuic acid; ROS, reactive oxygen species; SMACs, second mitochondria-derived activator of caspases.

**Table 1 antioxidants-08-00479-t001:** Biological functions of C3G-Ms.

C3G-Ms	Biological Functions	Objects	Results
PCA	Antioxidant	Rats, mice, macrophages	Treatment with PCA increased T-AOC [31], catalase [33], SOD [33] and GPx [33,34,35,36] levels, but decreased ROS [35], MDA [31] and hydroperoxides [31] levels.
	Anti-inflammatory	Mice, macrophages	PCA decreased IL-6 [33,37,39], TNF-α [33,39], IL-1β [33,39] and PGE_2_ production [39], and inhibited ERK, NF-κB p65 activation [33].
PGA	Anti-inflammatory	Mice, Human	PGA decreased serum levels of MCP-1 and TNF-α in high fat diet-induced mice [41]; PGA inhibited the production of IL-1β and IL-6 in human whole blood cultures after LPS stimulation, but no significant difference (*p* > 0.01) [40].
VA	Antioxidant	Hamsters, mice, rats	VA increased SOD [43,44], catalase [43,44], GPx [43,44], vitamin E [43,44], vitamin C [43,44] and GSH [43,44,45] levels.
	Anti-inflammatory	Rats, mice, macrophages	VA inhibited caspase-1, NF-κB and MAPKs activation [45,46,47], decreased production of COX-2, PGE_2_ and NO [46], and reduced the levels of TNF-α [45,46], IL-6 [46,55], IL-1β [45] and IL-33 [45].
FA	Antioxidant	Rats, mice, IEC-6 cells	FA decreased the production of ROS [45,46,47], MDA [49], NO [49], enhanced SOD [48,49] and CAT [48,51] activity, and promoted the activation of Nrf2 [51].
	Anti-inflammatory	HUVEC cells, mice, rats	FA decreased the expression of caspase-1 [52], ICAM-1 [52], VCAM-1 [52], IL-18 [52], IL-1β [50,52,53,54], IL-6 [50,54], TNF-α [53], and inhibited the phosphorylation of p38 and IκB [52].

Notes: C3G-Ms, cyanidin-3-glucoside metabolites; CAT, catalase; COX-2, cyclooxygenase-2; ERK, extracellular signal-regulated kinase; FA, ferulic acid; GSH, glutathione; ICAM-1, intercellular adhesion molecule-1; LPS, lipopolysaccharide; MAPKs, mitogen-activated protein kinases; MCP-1, monocyte chemoattractant protein-1; MDA, malondialdehyde; NF-κB, nuclear factor-κB; NO, nitric oxide; PCA, protocatechuic acid; PGA, phloroglucinaldehyde; PGE2, prostaglandin E2; ROS, reactive oxygen species; T-AOC, total antioxidant capacity; VA, vanillic acid; VCAM-1, vascular cell adhesion molecule-1; SOD, superoxide dismutase; TNF-α, tumor necrosis factor-α.

**Table 2 antioxidants-08-00479-t002:** Crosstalk between C3G&C3G-Ms and microorganism.

Microbial Species	Features	Bioconversion	Bacteriostasis
*Lactobacillus* (*L. paracasei*, *B. lactis* and *B. dentium*) and *Bifidobacterium*	Gram-positive anaerobes	C3G and cyanidin 3-rutinoside →PCA [60,61]	PCA─┤*E. coli*, *P. aeruginosa* and *S. aureus* [69]
*Lactobacillus* (*L. acidophilus K1* ) and *Bifidobacterium* (*B. catenulatum KD 14*, *B. longum KN 29* and *B. animalis Bi30*)	Gram-positive anaerobes	Methyl esters of phenolic acids →FA [57,62]	FA─┤*Botrytis cinerea* [71]
*Bacillus subtilis* and *Actinomycetes* (*Streptomyces sp. A3*, *Streptomyces sp. A5* and *Streptomyces sp. A13*)	Gram-positive facultative anaerobes	VA→guaiacol [63]	VA─┤*Pseudomonas* and *Bacillus spp.* [70]

Notes: →, generate; ─┤, inhibit.

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
