# Peer review of "The Effects and Mechanisms of Cyanidin-3-Glucoside and Its Phenolic Metabolites in Maintaining Intestinal Integrity"

_antioxidants, 2019, doi:10.3390/antiox8100479_

Round 1

Reviewer 1 Report

This is an interesting overview focusing primarily on the positive effects of Cyanidin-3-2 Glucoside and Its Phenolic Metabolites on anti-oxidant, anti-inflammatory and anti-apoptosis function in gut injury.

I think that the review is interesting, but in my opinion, there are some points that should be better investigated and organized. In particular:

First, I believe that figure 1 is confusing. I suggest giving more importance to the C3G molecule and its main metabolites in the intestine. Furthermore, I strong suggest improving the representation of the organs with a picture more similar to reality.

Paragraph 3:I believe that paragraph it should be summarized in a table. I suggest to indicate the biological function, the effect of the metabolites and the type of experimental model (in vitro, animal or human). It is well known that the alteration of the gut microbiota composition is at the basis of the development of numerous diseases. The authors in this review analyse the positive modulation of gut microbiota by C3G. How do they discuss the potential negative effects of this alteration?

The conclusions of this section do not seem to be new and original. Based on the contents of review, can you provide suggestions on natural anthocyanin use in clinical practice? I suggest widening the conclusion section with a more innovative comment.

Author Response

Response to Reviewer 1’s comments

This is an interesting overview focusing primarily on the positive effects of Cyanidin-3-2 Glucoside and Its Phenolic Metabolites on anti-oxidant, anti-inflammatory and anti-apoptosis function in gut injury.

I think that the review is interesting, but in my opinion, there are some points that should be better investigated and organized. In particular:

First, I believe that figure 1 is confusing. I suggest giving more importance to the C3G molecule and its main metabolites in the intestine. Furthermore, I strong suggest improving the representation of the organs with a picture more similar to reality.

Response: Thank you for your valuable suggestions. We improved Figure 1 for better understanding the metabolism of C3G in intestines with a better visual presentation (see Figure1).

According to a pharmacokinetics study of C3G in human (de Ferrars, R. M.; Czank, C.; Zhang, Q.; Botting, N. P.; Kroon, P. A.; Cassidy, A.; Kay, C. D., The pharmacokinetics of anthocyanins and their metabolites in humans. Br. J. Pharmacol. 2014, 171, 3268-82.), PCA, PGA, VA, FA and their derivates represent the main metabolites of C3G. Although hippuric acid is also one of the major metabolites, we paid little attention to it since it is considered as an end-product with limited biological activities. In addition, considering low concentration of C3G in plasma and high absorption of C3G in situ gastric perfusion model in rats (Talavéra S, Felgines C, Texier O, Anthocyanins are efficiently absorbed from the stomach in anesthetized rats. Journal of Nutrition, 2003, 133(12): 4178-4182.), it is believed that there was the first-pass metabolism for C3G in stomach (Fang Jim, Some Anthocyanins Could Be Efficiently Absorbed across the Gastrointestinal Mucosa: Extensive Presystemic Metabolism Reduces Apparent Bioavailability. Journal of Agricultural & Food Chemistry, 2014, 62(18): 3904-3911.). That is to say, C3G can be efficiently absorbed from the gastrointestinal lumen, undergo extensive first-pass metabolism, and enter the systemic circulation as metabolites.

The key steps of C3G metabolism in intestines can be concluded as follows:

Formation of aglycones by deglycosylation, which mainly happen in stomach and in proximal small intestine such as duodenum and jejunum; Cleavage of C-ring of C3G to form phage І metabolites such as protocatechuic acid (PCA) and phloroglucinaldehyde (PGA), which mainly happen in distal small intestine such as ileum; Dehydroxylation is to form phageⅡmetabolites such as vanillic acid (VA), ferulic acid (FA), PCA-3-glucuronide (PCA-3-Gluc), PCA-4-glucuronide (PCA-4-Gluc), PCA-3-sulfate (PCA-3-Sulf), PCA-4-sulfate (PCA-4-Sulf), VA-4-sulfate (VA-4-Sulf), isovanillic acid (IVA), IVA-3-sulfate (IVA-3-Sulf), which mainly happen in proximal large intestine such as colon; Decarboxylation is to form multistage metabolites including bacterial metabolites.

We added that information in the revise manuscript, either in text (Line 53-72) or in Figure 1.

Paragraph 3:I believe that paragraph it should be summarized in a table. I suggest to indicate the biological function, the effect of the metabolites and the type of experimental model (in vitro, animal or human). It is well known that the alteration of the gut microbiota composition is at the basis of the development of numerous diseases. The authors in this review analyse the positive modulation of gut microbiota by C3G. How do they discuss the potential negative effects of this alteration?

Response: Thank you for the suggestion. We added a table (Table 1) to summarize the antioxidant and anti-inflammatory activities based on different models in the revised manuscript. Yes, the alteration of the gut microbiota composition is at the basis of the development of numerous diseases. However, to our knowledge, most previous studies only highlighted the correlation between the beneficial effect and gut microbiota modulated by polyphenols. Only a few studies demonstrated that over-consumption of polyphenols had significantly negative effects on reproduction and pregnancy, and it is inexplicit whether there is a correlation with the modulation of gut microbiota. We are also focusing on the negative effects of polyphenols-modulated gut microbiota in our recent studies, but the results are not published yet. We made a brief discussion at the end of Chapter 4 to illustrate this shortcoming.

(Line 169-173): But it is worth noting that a few studies demonstrated that over-consumption of polyphenols had significantly negative effects on reproduction and pregnancy [72-74]. Although it is inexplicit whether there has a correlation with the changes of gut microbiota composition, the negative effects of polyphenols-mediated modulation of gut microbiota should be focused.

The conclusions of this section do not seem to be new and original. Based on the contents of review, can you provide suggestions on natural anthocyanin use in clinical practice? I suggest widening the conclusion section with a more innovative comment.

Response: Thanks for your suggestions. We improved the concluding remarks by extending the potential usage of anthocyanins in protecting against chronic diseases, the significance of maintaining intestinal integrity by anthocyanins, and the crosstalk between anthocyanins and gut microenvironment. We hope the improved conclusions will provide new ideas in understanding both the health benefits and negative effects of C3G and its metabolites in maintaining intestinal integrity (see line 330-357).

Reviewer 2 Report

The manuscript is valuable because the metabolic fate of many plant compounds after their consumption is not really clear. The Authors gathered data concerning this subject from very wide bibliography. The parts concerning the main subject are written properly and I don’t have bigger objections. However, in the introduction, some awkward expressions can be found which are spoiling the general impression exerted by the manuscript. Generally, introduction is the weakest part of this manuscript, a little chaotic and not precise. Below I suggest some changes which might improve the text.

Line 40.”widely existing in plants and fruits”. Fruits are parts of plants, it is very strange to say „plants and fruits” as we say „plants and animals”… I think that it should be just written: „widely occurring in plants”

and then, in line 41 continued as „ pigments contributing to the coloration of flowers and fruits.

Line 41. „within the plant” should be deleted.

Line 48. „naturally found in black rice, black bean, purple potato and other colorful berries” Why other?” Rice, bean, potato are not berries! So better: ”found in black rice, black bean, purple potato and many colorful berries”

Line 51. „have been identified in circulation”? I think it means: „blond circulation system”, or maybe just „in blood”?

Line 115. Bacteria can use phenolic compounds as a substrate to obtain energy, better „substrates” (in plural)

Line 199. „Potential mechanisms underlying C3G-Ms regulate Nrf2 system” = „regulated” would be more gramatically corrected

Line 307. Conventional wisdom has it that … Better simpler: It is commonly believed that…

Line 319 „there still are some problems need to be solved satisfactorily” – problems needed to be solved..

Author Response

Response to Reviewer 2’s comments

Comments:
”widely existing in plants and fruits”. Fruits are parts of plants, it is very strange to say „plants and fruits” as we say „plants and animals”… I think that it should be just written: „widely occurring in plants”

Response: Thank you for your kind and careful reviewing. We corrected it in the revised manuscript.

(Line 42): widely occurring in plants.

and then, in line 41 continued as „ pigments contributing to the coloration of flowers and fruits.

Line 41. „within the plant” should be deleted.

Response: We have corrected it. Thank you.

(Line 43): and fruits.

Line 48. „naturally found in black rice, black bean, purple potato and other colorful berries” Why other?” Rice, bean, potato are not berries! So better: ”found in black rice, black bean, purple potato and many colorful berries”

Response: We have already corrected it. Thank you!

(Line 50): many colorful berries.

Line 51. „have been identified in circulation”? I think it means: „blond circulation system”, or maybe just „in blood”?

Response: Thank you for your professional reminder. We have affirmed from original article that was “in serum”, and we corrected it.

(Line 53-54): in serum by a pharmacokinetics study in human.

Line 115. Bacteria can use phenolic compounds as a substrate to obtain energy, better „substrates” (in plural)

Line 199. „Potential mechanisms underlying C3G-Ms regulate Nrf2 system” = „regulated” would be more gramatically corrected

Line 307. Conventional wisdom has it that … Better simpler: It is commonly believed that…

Line 319 „there still are some problems need to be solved satisfactorily” – problems needed to be solved..

Response: Thank you for your kind reminder. All the shortcomings have been corrected. Thank you!

(Line128): substrates

(Line219): regulated

(Line 336-337): It is commonly believed that…

(Line 355): are still not clear.

Reviewer 3 Report

Cyanidin-3-glucoside (C3G) is one of the most common anthocyanins naturally found

in black rice, black bean, purple potato and other colorful berries, and possesses anti-oxidant and anti-inflammatory properties mainly through C3G metabolites (C3G-Ms). This paper gave a nice review on the effects and action mechanisms of C3G and its phenolic metabolites in maintaining intestinal integrity. A crosstalk between gut microbiota and C3G&C3G-Ms was presented. There are some concerns as listed in the following:

(1) Cite some of the following papers may strengthen the content of Part 4 (L114-)

*Food Chem Toxicol. 2019 Aug 23;133:110767. doi: 10.1016/j.fct.2019.110767. [Epub ahead of print]

Effects of cyanidin-3-O-glucoside on 3-chloro-1,2-propanediol induced intestinal microbiota dysbiosis in rats.

Chen G1, Wang G2, Zhu C1, Jiang X1, Sun J3, Tian L4, Bai W5.

*Food Res Int. 2019 Jun;120:523-533. doi: 10.1016/j.foodres.2018.11.001. Epub 2018 Nov 2.

Antioxidant potential and phenolic profile of blackberry anthocyanin extract followed by human gut microbiota fermentation.

Gowd V1, Bao T1, Chen W2.

*Eur J Nutr. 2005 Mar;44(3):133-42. Epub 2004 Apr 28.

In vitro metabolism of anthocyanins by human gut microflora.

Aura AM1, Martin-Lopez P, O'Leary KA, Williamson G, Oksman-Caldentey KM, Poutanen K, Santos-Buelga C.

*Bioorg Med Chem. 2005 Sep 1;13(17):5195-205.

Metabolism of anthocyanins and their phenolic degradation products by the intestinal microflora.

Keppler K1, Humpf HU.

R27*Br J Nutr. 2013 Apr 28;109(8):1433-41. doi: 10.1017/S0007114512003376. Epub 2012 Aug 21.

Contribution of gut bacteria to the metabolism of cyanidin 3-glucoside in human microbiota-associated rats.

Hanske L1, Engst W, Loh G, Sczesny S, Blaut M, Braune A.

(2) Typos and others

L95: Phaseâ…¡metabolties

*L109: IκB is not a kinase

L158-165: ??

L261: tight proteins?

L284: [101,102]. .

*L328: References: It should keep one format for the title (capital letter or not), journal name (abbreviation or full name).

Author Response

Response to Reviewer 3’s comments

Comments:
Cite some of the following papers may strengthen the content of Part 4 (L114-)

1.*Food Chem Toxicol. 2019 Aug 23;133:110767. doi: 10.1016/j.fct.2019.110767. [Epub ahead of print]

Effects of cyanidin-3-O-glucoside on 3-chloro-1,2-propanediol induced intestinal microbiota dysbiosis in rats.

Chen G1, Wang G2, Zhu C1, Jiang X1, Sun J3, Tian L4, Bai W5.

2.*Food Res Int. 2019 Jun;120:523-533. doi: 10.1016/j.foodres.2018.11.001. Epub 2018 Nov 2.

Antioxidant potential and phenolic profile of blackberry anthocyanin extract followed by human gut microbiota fermentation.

Gowd V1, Bao T1, Chen W2.

3.*Eur J Nutr. 2005 Mar;44(3):133-42. Epub 2004 Apr 28.

In vitro metabolism of anthocyanins by human gut microflora.

Aura AM1, Martin-Lopez P, O'Leary KA, Williamson G, Oksman-Caldentey KM, Poutanen K, Santos-Buelga C.

4.*Bioorg Med Chem. 2005 Sep 1;13(17):5195-205.

Metabolism of anthocyanins and their phenolic degradation products by the intestinal microflora.

Keppler K1, Humpf HU.

5.R27*Br J Nutr. 2013 Apr 28;109(8):1433-41. doi: 10.1017/S0007114512003376. Epub 2012 Aug 21.

Contribution of gut bacteria to the metabolism of cyanidin 3-glucoside in human microbiota-associated rats.

Hanske L1, Engst W, Loh G, Sczesny S, Blaut M, Braune A.

Response: We appreciate your kind reviewing. Yes, these papers are truly helpful to support our contents. We have already read Paper 2-5 and added them as references into our manuscript. However, we failed to access paper 1 (Effects of cyanidin-3-O-glucoside on 3-chloro-1,2-propanediol induced intestinal microbiota dysbiosis in rats.), might due to it is too new. Thank you!

L95: Phaseâ…¡metabolties

*L109: IκB is not a kinase

L158-165: ??

L261: tight proteins?

L284: [101,102]. .

Response: Thank you for your professional suggestion. We have corrected the ambiguous and mistakes in the revised manuscript.

(Line 99): Phase â…¡ metabolites of C3G

(Line 237): by degrading its inhibitory protein IκB

(Line 282): tight junction proteins

*L328: References: It should keep one format for the title (capital letter or not), journal name (abbreviation or full name).

Response: Thanks for your suggestion. All references have been corrected in the revised manuscript.

Round 2

Reviewer 1 Report

    Thanks for the careful and precise answers. I have no further comments.